# Optimized High Throughput Ascochyta Blight Screening Protocols and Immunity to *A. pisi* in Pea

**DOI:** 10.3390/pathogens12030494

**Published:** 2023-03-22

**Authors:** Emmanuel N. Annan, Bernard Nyamesorto, Qing Yan, Kevin McPhee, Li Huang

**Affiliations:** Department of Plant Sciences and Plant Pathology, Montana State University, Bozeman, MT 59717-3150, USA

**Keywords:** plant pathogens, fungi, pea, *Pisum sativum*, Ascochyta blight, high throughput screening protocols, immunity to Ascochyta blight

## Abstract

Ascochyta blight (AB) is a destructive disease of the field pea (*Pisum sativum* L.) caused by necrotrophic fungal pathogens known as the AB-disease complex. To identify resistant individuals to assist AB resistance breeding, low-cost, high throughput, and reliable protocols for AB screening are needed. We tested and optimized three protocols to determine the optimum type of pathogen inoculum, the optimal development stage for host inoculation, and the timing of inoculation for detached-leaf assays. We found that different plant development stages do not affect AB infection type on peas, but the timing of inoculation affects the infection type of detached leaves due to wound-induced host defense response. After screening nine pea cultivars, we discovered that cultivar Fallon was immune to *A. pisi* but not to *A. pinodes* or the mixture of the two species. Our findings suggest that AB screening can be done with any of the three protocols. A whole-plant inoculation assay is necessary for identifying resistance to stem/node infection. Pathogen inoculation must be completed within 1.5 h post-detachment to avoid false positives of resistance for detach-leaf assays. It is essential to use a purified single-species inoculum for resistant resource screenings to identify the host resistance to each single species.

## 1. Introduction

The Field pea (*Pisum sativum* L.) is an annual, cool-season legume cultivated across the globe. Field peas’ nutritional and environmental advantages to humans and animals cannot be overemphasized. The seeds of peas provide approximately 15–35% of the necessary amino acids, minerals, and carbohydrates [1,2,3]. Despite these benefits, pea productivity is threatened by biotic and abiotic stresses. Peas are susceptible to many bacterial, viral, and fungal pathogens, and pests of insects, mites, and nematodes [4].

Ascochyta blight (AB) is an important seed-borne and foliar disease of field peas and is widely recognized as a major productivity barrier. The disease is caused by necrotrophic fungal pathogens, *Ascochyta pisi* Lib. (*Didymella pisi* sp. nov), *A. pinodes* L.K. Jones (*D. pinodes*), *A. pinodella* L.K. Jones (*D. pinodella*), and *A*. *koolunga* (*Phoma koolunga*) [5]. These pathogens can exist independently or together (known as the Ascochyta blight disease complex) within a pea field and even on single plants [6,7]. *A. pinodes* receives particular attention across the globe due to its aggressiveness [7,8,9]. *A. koolunga* has not been reported in the Great Plains of North America [10].

During cool and damp weather, the pathogens can infect all parts of the plants [6,10,11,12], and reduce yield by 15~75% [13,14,15]. The disease progresses upwards from the bottom of the plant, characterized by lesions on leaves, stems, and pods that are tan, somewhat sunken, and sharply bordered by a prominent black line, with a chance of producing root rot [16]. The pathogens can over-winter on infected plant debris and in soil [8,17], and multiple infections can occur throughout the growing season, through the production and release of new spores during the wet periods.

There are regional variations in the pathogens causing AB diseases of field peas around the world [10]. *A. pinodes*, *A. pinodella*, and *A. koolunga* are the main pathogens linked to the disease in Australia [6]. *A. pisi* is the most prevalent pathogen in Europe and Asia, and was linked to recent disease outbreak in Georgia and Spain [18,19]. According to a field survey of Montana’s field pea growing regions and seed tests at the Regional Pulse Crop Diagnostic Laboratory (RPCDL) in 2017, *A. pisi* was also found to be the most common species in Montana, while *A. pinodes* was up to 5% of the fungal species isolated from plants with Ascochyta blight and contaminated seed lots [10]. However, the presence of *A. pinodes* and *A. pinodella* in North Dakota and Saskatchewan [13,20,21] raises serious concerns that these aggressive species might gradually spread to and predominate in Montana because the fungal spores may transcend by wind through geographically closed field pea producing areas of Saskatchewan and North Dakota to Montana in addition to regular seed exchange among these regions. A study by Tivoli & Banniza (2007) showed that once *A. pisi* resistance was introduced in the germplasm used in Canada, *A. pinodes* became the predominant species in the regions. Ascochyta blight pathogen population has been dynamically changed in Western Australia to the point that *A. koolunga* has replaced *A. pinodes* in the field pea. Sustainable management of pea AB disease must target all species of the complex.

The current strategies for managing Ascochyta blight include using blight-free seeds, seed treatment, crop rotation, fungicide applications [22,23,24], and host resistance. The most environment-friendly and cost-effective method of managing pea Ascochyta blight would be the use of resistant cultivars. To date, all reported sources of resistance identified exhibited only modest resistance against some but not all species of the Ascochyta complex [12,25,26,27]. The lack of effective host resistance becomes the bottle-neck of resistant cultivar breeding [23]. To assist phenotyping to find more resistance resources and selection for AB-resistance breeding programs, a high throughput and reliable AB screening protocol is essential.

To date, there are two commonly used protocols to assess AB disease severity: whole-plant spray inoculation and detached-leaves pipette inoculation [28,29,30,31]. A whole-plant assay provides an opportunity to examine the progress of blight as well as its severity in different plant organs, including leaves, stems, and nodes. However, the pathogens could kill the host before it bears seeds. Detached-leaf pipetted-inoculation has its advantages in tracking and measuring lesion sizes and saving the plants for seeds although it is time consuming and labor-intensive, and blight symptoms are only examined on leaves. Further, the wounding caused by detaching can activate the plant defense response which may lead to false positives of resistance. The purpose of our study is to optimize the current protocols for high throughput and reliable AB screenings to identify new resistance to AB pathogens.

## 2. Materials and Methods

### 2.1. Plant Materials and Growth Conditions

Nine spring pea varieties, including “Fallon”, “Hampton”, “DS Admiral”, “Agassiz”, “CDC Treasure”, “SW Arcadia”, “CDC Amarilla”, “Greenwood”, and “Aragon” were used in this study.

Pea plants were grown in a greenhouse with a day/night temperature of 30.2/18 °C, 22% humidity and 16 h of light at the Plant Growth Centre of Montana State University (MSU). The seeds were sown in soils containing a mixture of peat and sunshine mix 1 (Sun Gro Horticulture, Agawam, MA, USA) at a ratio of 1:1.

### 2.2. Pathogens and Growth Conditions

Two AB fungal species were used for the study; they are *A. pinodes* isolate “C6” and *A. pisi* isolate “22-P2033”. The pathogens were isolated from AB-contaminated seed lots submitted by growers to the Regional Pulse Crop Diagnostic Laboratory (RPCDL) at MSU. The two fungal species were purified and each was confirmed as a pure culture of the species by PCR using species-specific primers [32] and sequencing. The isolates were cultured on a pea meal agar, consisting of 25 g of blended dry pea and 15 g of Agar in 1 L of double-distilled water (ddH_2_O), at room temperature for the reproduction of conidia.

### 2.3. Inoculum Preparation and Inoculation

Single-species inocula contained only a pure species of either *A. pinodes* or *A. pisi*, were prepared by flooding the cultured surface with 30 mL sterilized ddH_2_O and gently scraping the flooded culture surface with a glass rod for 10 min to dislodge conidia. Conidia spores were then collected after filtration through one layer of cheesecloth and used as stocks. The spore density of the stocks was counted with 10 µL pipetted onto a hemacytometer under a light microscope. The inocula were prepared by diluting a calculated amount of each stock with sterilized ddH_2_O + Tween 20 (two drops per 200 mL) to achieve a spore density of 1 × 10^5^ spores/mL.

Mixed-species inocula in this study refer to the inocula with an equal number of spores of *A. pinodes* and *A. pisi* but prepared in two ways. The first way was to combine an equal volume of the two single-species inocula prepared and used as a mixed-species inoculum. In this inoculum, the spore density of each species was equivalent to 50% of the spores in the single-species inoculum. The second was to take a calculated amount of each stock of single-species and dilute it to 1 × 10^5^ spore/mL/species, so the spore density of each species in the mixture was equivalent to that of the single-species inoculum, but the total spore density in the inoculum was doubled to 2 × 10^5^ spores/mL.

For the whole-plant assay, an inoculum was applied to plants by spraying (Figure 1a). For the detached-leaf assay, the second node stipules were used. The two connected pea stipules were separated and placed upside into two cells of the same column of a 6-well plate floated on ddH_2_O. Then, the same inoculum was either spray inoculated (Figure 1b) or pipette (10 µL/stipule) inoculated onto the stipules (Figure 1c). In the spraying process, we ensured that every part of a plant was covered with the inoculum. The inoculated plants were kept in a dew chamber with an air temperature of 22 °C for 24 h in the dark and then transferred to a greenhouse and kept inside a plastic tent with mists produced by humidifiers for at least 12 h.

### 2.4. RNA Extraction and PR4 Gene Expression Assay

Leaf samples were frozen in liquid nitrogen and kept in a −80 °C refrigerator if the tissues were not used immediately. RNAs were isolated and treated with DNase I in the process of extraction on columns using the Qiagen RNeasy Plant Mini Kit (Qiagen, Valencia, CA, USA) based on the manufacturer’s instructions. The concentration of total RNA was assessed using 260/280 ABS measurements on a NanoDrop 1000 spectrophotometer (Thermos Fisher Scientific Inc., Wilmington, DE, USA). The integrity of the RNAs were checked via agarose gel electrophoresis with 1 μL of the sample, 4 μL of water, 1 μL loading buffer (98% formamide, 10 mM EDTA, 0.25% bromophenol blue, and 0.25% xylene cyanole) on a 1% gel stained by GelRed (Bio-Rad, Hercules, CA, USA) at 120 volts for 30 min.

Gene expression was measured as relative expression to a reference gene by quantitative real-time (qRT-PCR) performed in a CFX96 real-time PCR detection system (Bio-Rad, Hercules, CA, USA). The PCR was set up using the iScript One-Step RT-PCR Kit with SYBR Green (Bio-Rad, Hercules, CA, USA) in a 20 μL total reaction mix volume containing 2 μL of a mixture of forward and reverse primer (1 pmol/μL), 10.25 μL of SYBR Premix, 2 μL of RNA (37.5 ng/μL), and 5.75 μL of RNase-free water. RNA at annealing temperatures of 55 °C. Protein Phosphatase 2A was used as the reference gene for calculating relative transcript abundance of *PR4* because of its stability under biotic and abiotic conditions [33]. Primer sequences for qRT-PCR are listed in Table 1. The *PR4* gene expression was presented as a fold change of 2^−ΔΔCt^. Three biological replications were tested, and three technical repeats were set for each reaction.

### 2.5. Statistical Analysis

Statistical analysis was performed using R-software (R-4.2.0). All the experiments were conducted in controlled conditions with two factors, cultivar and inoculum. Disease severity was documented as number of dead leaves/plant and infected nodes/plant. The data were analyzed using one-way ANOVA (analysis of variance). Mean values of disease severity of each species were calculated for each cultivar. Mean values and box plots were generated for each cultivar against each species and visualized using ggboxplot in R software. Statistical differences between mean values were determined by a one-way ANOVA and Tukeys’ HSD.

## 3. Results

### 3.1. Optimization and Comparison of the High Throughput Protocols

A high throughput protocol should be able to handle a large number of samples at a given time and to obtain results within a short period of time. The inoculation method is the most critical parameter for a disease screening protocol in determining sample-testing capacity. As for choosing current whole-plant spray inoculation or detached pea stipules and pipetted-inoculation of pathogen conidia spores as protocols in Ascochyta blight screening each has its advantages and disadvantages. First, we assessed three inoculation protocols to test if different assays could reveal comparable results on the same pea variety. A single-species inoculum of *A. pinodes* was tested on one variety, “Fallon” (Figure 1 and Figure 2), with the three assays. Visible symptoms were first observed on the inoculated stipules at 24 h post-inoculation (hpi). At 10 dpi, spray-inoculated stipules on the whole plant (Figure 2a) or detached (Figure 2b) were dead, and enlarged dead brown spots appeared on some of pipette-inoculated stipules (Figure 2c). Newly appearing stipules post inoculation on the whole plants were healthy and symptom-free (Figure 2a). However, if stem infections occurred, this plant could collapse at 20 dpi (Figure 2a). The three assays revealed the same result that Fallon was susceptible to *A. pinodes*. Apparently, spray-inoculations produced more uniform and consistent symptoms than those with pipette-inoculation (Figure 2).

The next step was to determine if the timing of inoculation for detach-leaf assays affects the reliability of the results. We want to know if wound-induced response changes the AB infection type on peas. Three pairs of stipules from the variety Fallon were inoculated with the same *A. pinodes* inoculum at 0, 1.5 and 4 h post-detachment (hpd). The results showed that the stipules at 4 hpd showed a false positive of resistance, while the stipules from 0 and 1.5 hpd were susceptible to the pathogen (Figure 3a).

### 3.2. Expression Profiles of PR4 in Detached Leaves

We hypothesized that wound-induced response enhanced resistance to AB pathogens. To test this hypothesis, we measured the expression of a wound-induced pathogenesis-related gene 4 (*PR4*) in detached stipules at 0, 1.5, and 4 hpd using qRT-PCR. The relative expression of *PR4* at 1.5 hpd was upregulated, but not significantly, compared to the level at 0 hpd (Figure 3b), which was consistent with the infection type (Figure 3a). *PR4* abundance was about 7.5-fold higher at 4 hpd than the level at 0 hpd (Figure 3b), which supported our hypothesis.

### 3.3. Different Development Stages Do Not Affect Pea Infection Type to AB

Another feature of a high throughput protocol is to obtain results within a short period of time. To do so, we need to determine the best developmental stage of a pea plant for pathogen inoculation. The thought behind this experiment was to determine the earliest stage of inoculation that can be undertaken to save time and space. Fallon seeds were sown seven days apart, so we could inoculate the plants at two different developmental stages with the same fungal inoculum simultaneously, while they are all kept under the same optimum conditions. The *A. pinodes* inoculation was done on 10-day-old plants with four nodes and six fully expanded stipules (Figure 4a) and on 17-day-old plants with six nodes and ten stipules (Figure 4b). There were no differences in disease progression among the stages of plant maturity (Figure 4). At nine dpi, six stipules from the 4-node stage plants (Figure 4a) and ten from the 6-node stage plants (Figure 4b) were all dead. We did not observe significant differences in disease severity associated with the plants’ age; the level of damage caused by the fungi in a 10-day-old plant (Figure 4a) was similar to that of a 17-day-old plant (Figure 4b). This result suggested that fungal inoculation can be done at any stage as long as the plants have fully expended stipules to catch the inoculum.

### 3.4. Host Responses to the Pathogens

After establishing the primary parameters of the protocol, the experiment was taken further by testing eight additional varieties and another pathogen, *A. pisi*. Moving forward, whole plant sprayed-inoculation was the preferred method due to its effortless application, and the availability of enough seeds for the experiment. These experiments served two purposes. One was to test if the protocol could produce repeatable results. The next was to identify resistance to each of the fungal species of the Ascochyta complex. Ten plants from every variety, including Fallon, were treated at the 6-node stage with *A. pinodes* and *A. pisi*, respectively. Inoculated plants were visually assessed on the number of dead leaves & infected nodes per plant. The experiments produced repeatable results on Fallon with the same level of disease severity with *A. pinodes* inoculation. The nine cultivars had no significant differences in the number of dead leaves (*p* = 0.13; Figure 5a) after treatment with *A. pinodes,* but the number of infected nodes per plant were significantly different (*p* = 6 × 10^−4^; Figure 5b). Hampton had the lowest mean average, while Fallon, Aragon, and Greenwood had a high average. Furthermore, all nine varieties had stem infections. To conclude, none of the nine cultivars showed sufficient levels of resistance against *A. pinodes*.

*A. pisi* had a similar growth rate as *A. pinodes* on pea meal agar during culture; however, the pathogenesis of *A. pisi* on susceptible pea plants was much slower and leaf blight severity was less compared to *A. pinodes*. Blight symptoms caused by *A. pisi* first appeared on some pea varieties at 6 dpi, and it took only 24 h for *A. pinodes* to cause pea leaf blight. *A. pisi* did not cause any visible symptoms on Fallon (Figure 6a) but caused different levels of blight on the rest of the eight varieties, for example, Greenwood (Figure 6b) and DS Admiral (Figure 7b). Notably, the numbers of dead leaves/plant were significantly different, although the eight varieties all had leaf blight (Figure 8a). None of the Agassiz infected leaves died, and Hampton had the highest mean of dead leaves/plant (*p* = 1.4 × 10^−5^, Figure 8a). Significant differences were also observed when the number of infected nodes per plant was analyzed (*p* < 2.2 × 10^−16^; Figure 8b). Excluding Fallon, all eight cultivars showed blight symptoms in the stems and nodes. CDL Treasure had the lowest mean average, while SW Arcadia, Greenwood, and Aragon had the highest mean averages.

Contrary to the results of *A. pinodes* inoculation, different levels of resistance to *A. pisi* were observed among the nine cultivars, and Fallon was immune to this pathogen. Another difference we noticed was that, although many fewer leaves died from the infection, node and stem infections were observed among eight cultivars. The plants with either node or stem infection could eventually lodge and die, so *A. pisi* infection should not be overlooked.

As in nature the two pathogen species often co-exist. We asked two further questions after our finding of Fallon’s resistance to *A. pisi*. 1. Does an *A. pisi* inoculation induce a sufficient level of systemic acquired resistance (SAR) against *A. pinodes*? 2. What is Fallon’s response to a mixture of *A. pisi* and *A. pinodes*? To answer the first question, we inoculated Fallon with *A. pinodes* at eight days post-*A. pisi* inoculation (Figure 9), and then monitored disease development and severity post-*A. pinodes* inoculation. Pretreatment with *A. pisi* did not slow the leaf blight development and severity (Figure 9). Furthermore, we conducted *A. pinodes* inoculation on the plants at 24, 48, and 72 h post- pre-*A. pisi* inoculation. This experiment was based on the knowledge that SAR could be induced just hours after avirulent pathogen infection. There was no significant difference in disease severity among different time points of pre-*A. pisi* treatment (Figure 10a,b), suggesting no sufficient levels of SAR induced by a pre-*A. pisi* treatment in pea against *A. pinodes*.

To determine if resistance to *A. pisi* in a host could enhance its resistance to *A. pinodes* in the complex, we used two types of mixed-species inocula (for details, see M&M). We compared mixed species-inoculum A (50% *pinodes* spores to its *pinodes*-A only inoculum); significant differences were observed between the number of dead leaves/plant (*p* = 3.1 × 10^−6^; Figure 11a) and infected nodes/plant (*p* = 1.2 × 10^−13^; Figure 11b), suggesting less disease severity when the density of *A. pinodes* spores was reduced. The finding was confirmed when we compared the plants with a mixed-species inoculum B and *pinodes*-B only inoculum when both inocula contain the same density of *A. pinodes* spores. There was no difference between both categories of dead-leaves/plant and infected nodes/plant (Figure 11a,b).

## 4. Discussion

### 4.1. Optimized Ascochyta Blight Inoculation Protocols

An important goal of this work was to develop low cost, rapid and reliable methods for screening large numbers of field pea genotypes to assist in AB-resistance breeding. Each of the three assessed methods tested in this study has their advantages and disadvantages. With a whole-plant inoculation assay, we were able to evaluate AB infection on various parts of a plant. Besides, whole-plant spray inoculation is easier to undertake, requires no technical know-how and is time-efficient. Since inoculation is applied as a fine mist, hundreds of plants could be inoculated with one pass of inoculation. Inocula can be uniformly applied to every plant to avoid false-positive-resistance due to a skip of inoculum. This was confirmed in our study, where we consistently noticed uniform blight symptoms 24 h after inoculation. AB symptoms appeared on every susceptible part of the plant with whole-plant inoculation assay. However, this assay is not suitable if offspring of the plants are needed for further analysis, such as genetic analysis or resistance gene mapping, because AB-infected plants likely die before they produce seeds. An alternative to high throughput AB screening is detached-leaf spray assay. Detached-leaf pipette-inoculation assay is useful in measuring disease lesion expansion when quantitative characterization of the symptoms is needed for a small population. We are aware that detached leaves are wounded, and wound-mediated response could enhance resistance to necrotrophic pathogens. For example, Ethylene and wound signaling were found to provide resistance to *Botrytis cinerea* in tomato [34]. From our investigation, wound-induced *PR4* had upregulated at 1.5 hpd, although no symptom changes were observed. We recommend that the pathogen inoculation must be completed within 1.5 h post-detachment.

We also investigated the stage of plant development optimum for pathogen inoculation. The developmental stage of genotypes is likely to influence disease outcome [35,36] with early emergence plants expected to be more or less susceptible. No difference in AB disease development was observed when plants were inoculated at different stages of maturity, and in our case, plants were sown seven days apart. Thus, we recommend that pathogen inoculation can be undertaken as early as ten days after sowing for a fast turnout of results and increasing screening capacity.

We conclude that the three AB screening protocols could produce comparable results if the recommendations are followed. The choice of method should be based on the scientific aims, but a whole-plant inoculation assay is necessary for identifying resistance to stem/node infections.

### 4.2. Different Pathogenicity Factors by Different Species in the Ascochyta Complex

Our study discovered that Fallon was immune to *A. pisi* but fully susceptible to *A. pinodes*. Furthermore, the immune response to *A. pisi* did not translate to any detectable enhanced resistance to *A. pinodes* when the two species co-exist, suggesting the immunity is species-specific and these pathogens deploy distinct pathogenicity factors.

Histology investigation on lesions on different parts of infected pea plants suggested phytotoxic compounds released by the pathogen were the main cause since plant cells were damaged and disintegrated before direct contact with the fungus [37]. The primary phytotoxic compounds produced by *A. pisi* and *A. pinodes* are different. The phytotoxic compound from *A. pisi* culture extracts is ascochitine, an o-quinone methide, causing electrolyte leakage on susceptible cultivars when a purified form of the compound is directly applied to leaf discs [38]. In contrast, *A. pinodes* produces pinolidoxin, a macrolide having a 10-member structure and a medium-sized lactone ring exhibiting strong phytotoxicity [39,40,41] that inhibits the plant phenylpropanoid pathway [42]. The phenylpropanoid compounds provide plants with preformed or induced physical and chemical barriers against infection [43]. Thus, the mode of pinolidoxin’s action is likely to interfere with the plant’s defensive responses during hyphal invasion [44].

These findings have a significance in new AB resistance identification and suggest the importance of using a purified single-species inoculum for resistance screenings. It will be impossible to identify host resistance if the inoculum contains more than one species of the AB pathogens.

### 4.3. AB Resistance Breeding in Pea Requires Gene Pyramiding

The genetics of AB resistance in different crop legumes have been disclosed by classical genetic investigations, supported by evidence of either monogenic inheritance regulated by a single dominant gene, a single recessive gene, or digenic [44] and quantitative inheritance [45,46]. Besides the immune response by Fallon, the other eight pea varieties tested at the same time showed different levels of resistance/susceptibility to *A. pisi*, suggesting the resistances are either qualitative and/or quantitative in different pea lines. Inheritance and nature of the Fallon immunity to *A. pisi* require further genetic analysis and gene cloning.

Our studies revealed that peas respond to AB species not only in specificity but are also spore-density dependent. Disease severity was significantly less when the spore density of *A. pinodes* in the mixed-species inoculum was reduced to 50% of the *A. pinodes* only inoculum. These findings suggest that phenotyping AB resistance in field trials must reference the compositions and density of each AB species in the locations. Since resistance to one AB species does not confer resistance to other species of the complex, AB resistance breeding requires resistance to each species and pyramiding of all the resistance to achieve a sufficient level and longevity of a resistant variety. Although the Ascochyta blight pathogen population showed regional patterns, the dynamical changes of *A. koolunga* replaced *A. pinodes* in the field pea in Western Australia, suggesting that sustainable management of pea Ascochyta blight disease must target all species of the complex.

## Figures and Tables

**Figure 1 pathogens-12-00494-f001:**
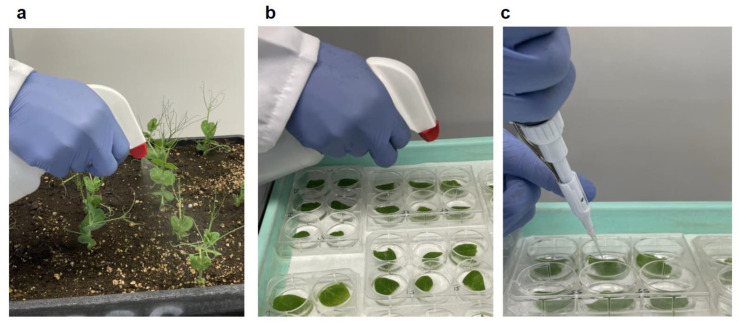
Three different inoculation protocols. Preparation of inocula was the same with a final concentration of 1 × 10^5^ spores/mL. (**a**) The whole-plant assay had inoculum applied to plants by spraying. (**b**) Detached-leaf assay with inoculum applied by spraying. (**c**) Detached-leaf assay with 10 µL of inoculum applied by pipetting.

**Figure 2 pathogens-12-00494-f002:**
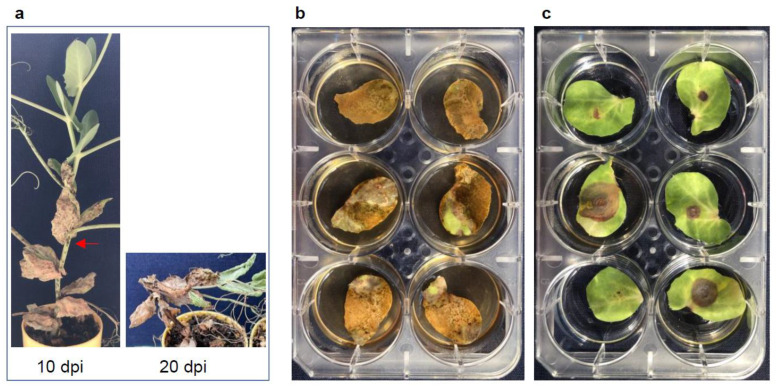
Response of Fallon to *A. pinodes* infection with the three inoculation methods. The same pure culture of *A. pinodes* (C6) and cultivar Fallon were used with the three assays. (**a**) Infection type with whole plant assay at 10 dpi and lodging at 20 dpi due to the stem infection. The red arrow shows stem infection. (**b**) Infection type with detached-leaf spray assay at 10 dpi, (**c**) Infection type with detached-leaf pipetting assay at 10 dpi.

**Figure 3 pathogens-12-00494-f003:**
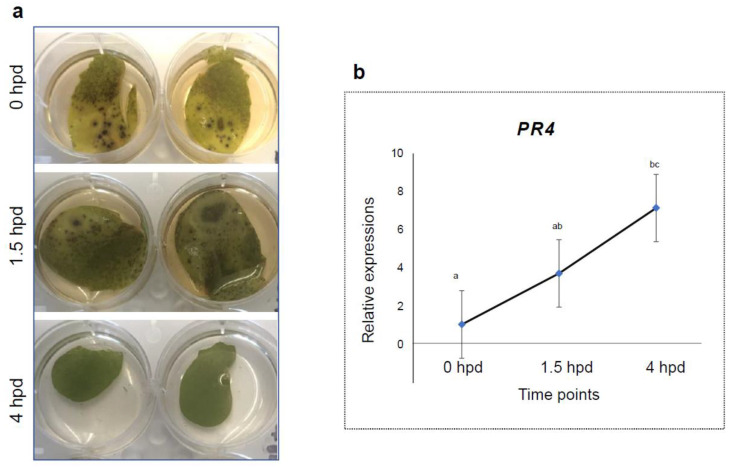
Wound-induced response and effect on the infection type on Fallon to *A. pinodes*. (**a**) Infection type of detached leaves at 10 days post-inoculated with the same *A. pinodes* inoculum at 0, 1.5 and 4 h post detachment (hpd). (**b**) Relative expression of *PR4* in detached leaves at 0, 1.5 and 4 hpd. Expression of *PR4* genes at each time point was computed relative to the level at 0 hpd. Error bars represent standard deviation and represented letters denote statistical significance at the *p* ≤ 0.05 levels calculated between each time point compared with 0 hpd.

**Figure 4 pathogens-12-00494-f004:**
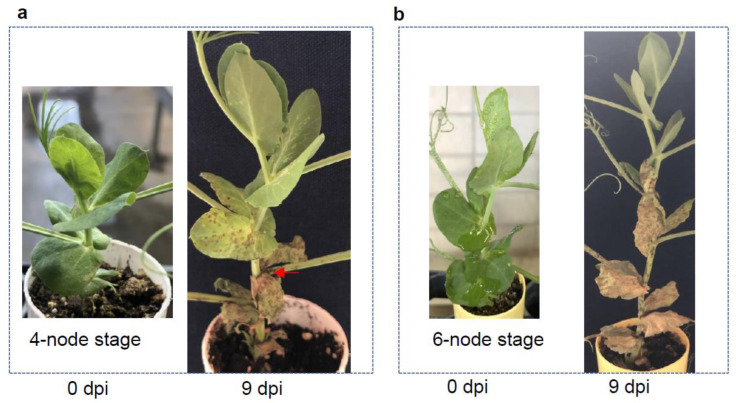
Infection types of Fallon inoculated at two developmental stages with the same *A. pinodes* inoculum. (**a**) Fallon was inoculated at 4-node stage and the infection type at 9 dpi. (**b**) Fallon was inoculated at 6-node stage and the infection type at 9 dpi. The red arrow shows node infection. The dpi under each plant indicated the time point when the photo was taken.

**Figure 5 pathogens-12-00494-f005:**
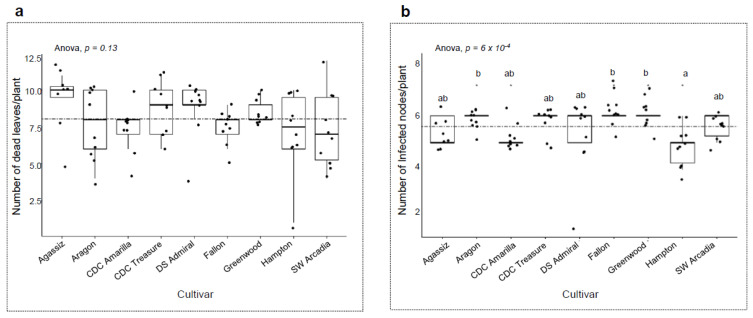
Severity of disease on nine varieties with *A. pinodes* inoculation. The inoculum was whole-plant spray inoculated on to the plants at the 6-node stage. Ten plants per variety were measured based on number of dead leaves and infected nodes per plant. Means of variety were compared using Anova and *p* ≤ 0.05 for pairwise comparison (Tukey’s HSD) against all varieties. (**a**) Means of number of dead leaves per variety. (**b**) Means of number of infected nodes per variety. The stippled line in each figure represents the mean of all data points accordingly. The letters in the figures indicate the significant levels among the cultivars, the same letter means no significant difference, different letters indicate significant levels < 0.05. “*” in the figures indicate the significant levels of ≤0.05.

**Figure 6 pathogens-12-00494-f006:**
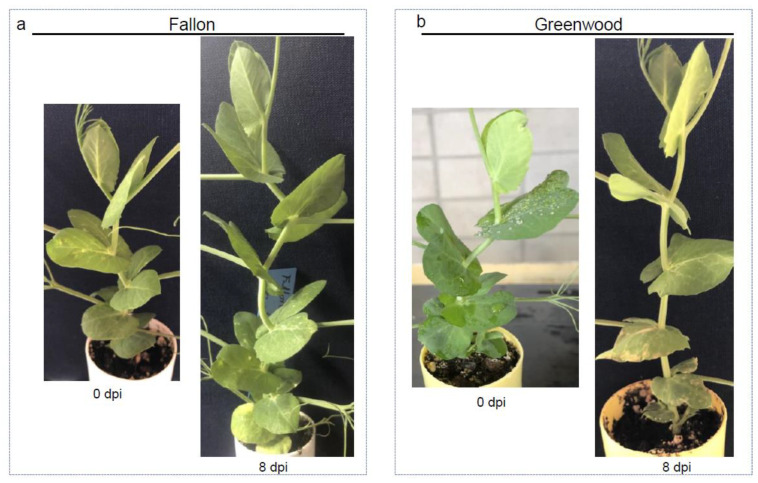
Infection types of Fallon and Greenwood with *A. pisi* inoculum. The two varieties were spray inoculated with the same pure cultures of *A. pisi* (22-P2033). (**a**) Fallon was immune to *A. pisi*. (**b**) Greenwood showed some level of susceptibility to *A. pisi*. The dpi under each plant indicated the time point at which the photo was taken.

**Figure 7 pathogens-12-00494-f007:**
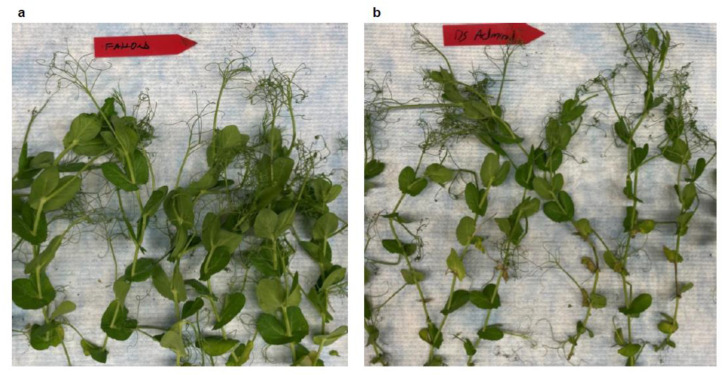
Infection types of Fallon and DS Admiral to *A. pisi*. The varieties were spray inoculated with pure cultures of *A. pisi* (22-P2033) and pictured at 10 dpi. (**a**) Fallon was immune to *A. pisi*. (**b**) Ds Admiral was susceptible to *A. pisi*.

**Figure 8 pathogens-12-00494-f008:**
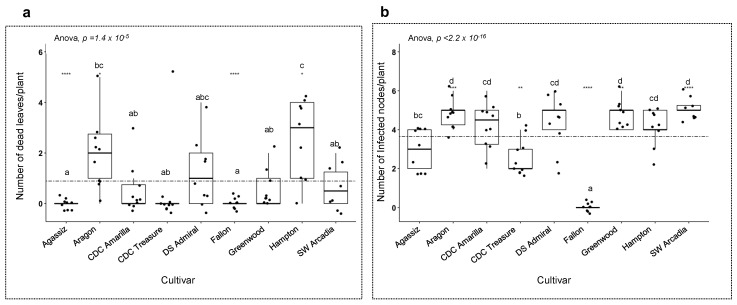
Severity of disease on nine varieties with *A. pisi* inoculation. Nine varieties were spray inoculated with *A. pisi* inoculum at the 6-node stage. Ten plants per variety were measured based on the number of dead leaves and infected nodes per plant. Means of varieties were compared using Anova and *p* ≤ 0.05 for pairwise comparison (Tukey’s HSD) against all varieties. (**a**) Anova comparison of means of the number of dead leaves per variety. (**b**) Anova comparison of means of the number of infected nodes per variety. The stippled line in each figure represents the mean of all data points accordingly. The letters in the figures indicate the significant levels among the cultivars, the same letter means no significant difference, different letters indicate significant levels <0.05. “****”, “***”, “**” and “*” in the figures indicate the significant levels of ≤0.0001, ≤0.001, ≤0.01 and ≤0.05, respectively.

**Figure 9 pathogens-12-00494-f009:**
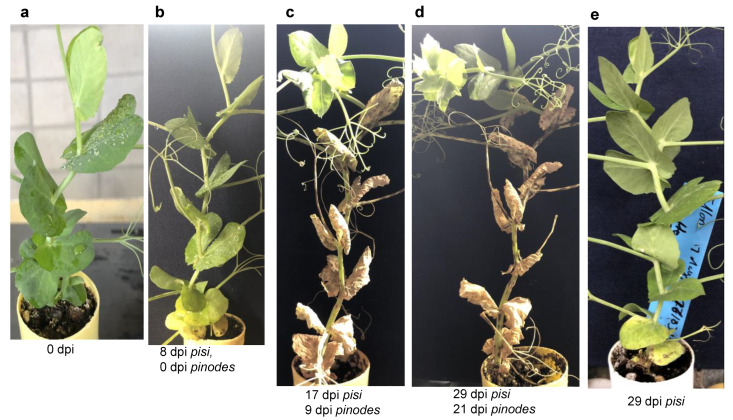
Infection types of Fallon to *A. pinodes* post induction of systemic acquired resistance by *A. pisi*. Plants were first spray inoculated with *A. pisi*, followed by another spray inoculation with *A. pinodes* 8 days later. (**a**) Fallon was inoculated with *A. pisi* at 6-node stage. (**b**) Infection type of Fallon to *A. pisi* at 8 dpi, and then inoculated with *A. pinodes. (***c**) Infection type of the plant after the double inoculation at 9 dpi with *A. pinodes,* 17 dpi with *A. pisi. (***d**) Infection type of the plant after the double inoculation at 21 dpi with *A. pinodes,* 29 dpi with *A. pisi. (***e**) Infection type of a plant with only *A. pisi* inoculation as a control. The dpi under each plant indicated the time point at which the photo was taken.

**Figure 10 pathogens-12-00494-f010:**
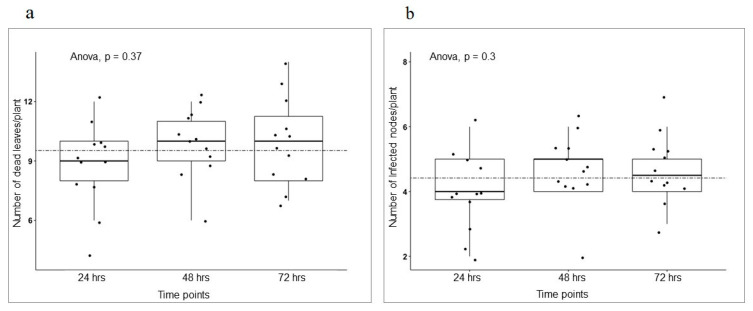
Severity of disease on Fallon with the possibility of SAR against *A. pinodes*. Plants were spray inoculated with *A. pisi*, followed by inoculation with *A. pinodes* at 24, 48 and 72 h. Ten plants per time point were measured based on dead leaves and the number of infected nodes per plant. Means of time points were compared using Anova, and *p* ≤ 0.05 for pairwise comparison (Tukey’s HSD) against all varieties. (**a**) Anova comparison of means of the number of dead leaves/plant/time point. (**b**) Anova comparison of means of the number of infected nodes/plant/time point. The stippled line in each figure represents the mean of all data points accordingly.

**Figure 11 pathogens-12-00494-f011:**
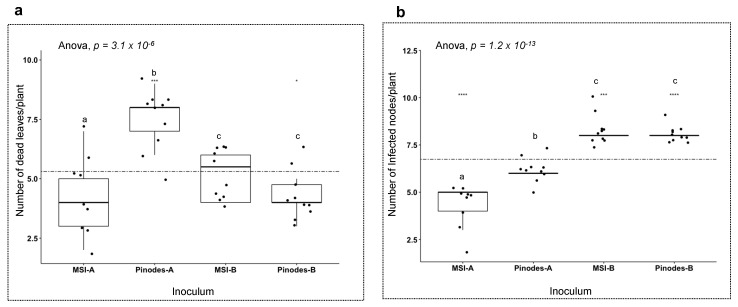
Severity comparison of Fallon between single-species and mixed-species inocula. Ten plants per treatment; MSI-A (50% *A. pisi* and 50% *A. pinodes* equivalent to 1 × 10^5^ mixture-species inoculum), MSI-B (100% *A. pisi* and 100% *A. pinodes* equivalent to 2 × 10^5^ mixture-species inoculum), *pinodes*-A, and pinodes-B. MSI-A inoculum is 50% *A. pinodes* spores to *pinodes*-A only inoculum, and MSI-B inoculum is 100% *A. pinodes* spores to *pinodes*-B only inoculum. Means of 10 plants were compared using Anova and *p* ≤ 0.05 for pairwise comparison (Tukey’s HSD). All inoculations were spray inoculated. (**a**) Comparison of means of the number of dead leaves per treatment. (**b**) Comparison of means of the number of infected nodes per treatment. The stippled line in each figure represents the mean of all data points accordingly. The letters in the figures indicate the significant levels among the cultivars, the same letter means no significant difference, different letters indicate significant levels < 0.05. “****”, “***” and “*” in the figures indicate the significant levels of ≤0.0001, ≤0.001 and ≤0.05, respectively.

**Table 1 pathogens-12-00494-t001:** Primer sequences.

Name	Sequence	Tm (°C)
PsPR4F	GCTGGGACTTAAACGCTGTT	55
PsPR4R	TGTGCTCCTGTCCCTGAATT	56
PP2AF	CCACATTACCTGTATCGGATGACA	55
PP2AR	GAGCCCAGAACAGGAGCTAACA	57

## Data Availability

The data presented in this study are available on request from the corresponding author.

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
