# Peer review of "Optimized High Throughput Ascochyta Blight Screening Protocols and Immunity to *A. pisi* in Pea"

_pathogens, 2023, doi:10.3390/pathogens12030494_

Round 1

Reviewer 1 Report

This interesting manuscript provides new methodological strategies to study Ascochyta blight in large experimental settings. I have only one main concern, which is related to the fungal isolates. Do they have the same growth rate? this parameter should be studied and included in the manuscript. The authors compare the host's response against the two species; and in the cases where differences were observed these could be due to differences in growth rate and thus adaptation of the host milieu. This is more relevant when combining the two fungal species to challenge the host. If one grows faster than the other, the results could be biased.

As a minor but relevant point, the described growth conditions are for conidia production, not spores. Please modify the manuscript accordingly.

Author Response

We appreciate the comments from the two reviewers that helped to improve the manuscript. Our responses to each question of the reviewer are followed.

Reviewer 1’s Comments and Suggestions for Authors

This interesting manuscript provides new methodological strategies to study Ascochyta blight in large experimental settings. I have only one main concern, which is related to the fungal isolates. Do they have the same growth rate?

--Response: The two species of Ascochyta pisi and A. pinodes had a similar growth rate on pea meal agar during culture. We added this information in the revised manuscript.

this parameter should be studied and included in the manuscript. The authors compare the host's response against the two species; and in the cases where differences were observed these could be due to differences in growth rate and thus adaptation of the host milieu. This is more relevant when combining the two fungal species to challenge the host. If one grows faster than the other, the results could be biased.

--Response: We appreciate the suggestion and a study to look into the interactions between the two species in susceptible pea plants is currently ongoing. We hope to convince the reviewer that this manuscript is about high throughput screening protocols and Fallon’s resistance to A. pisi. We did not present the study of co-inoculation of the two species onto any susceptible varieties. Since “Fallon” is immune to A. pisi, we anticipate no pathogen growth in “Fallon”. Our results showed that this immunity did not prevent A. pinodes infection on Fallon.

As a minor but relevant point, the described growth conditions are for conidia production, not spores. Please modify the manuscript accordingly.

--Response: Thank you for the suggestion! The M&M section of the manuscript has been modified accordingly.

Reviewer 2 Report

The manuscript generally reads well, pictures are nice and helpful, and relevant for this important topic.

You use the following taxonomy: Ascochyta pisi Lib. (Didymella pisi sp. nov), Peyronellaea pinodes 35 (Berk & A. Bloxam) Aveskamp, Gruyter & Verkley 2010 (Mycosphaerella pinodes (Berk & 36 A. Bloxam) Vestergr. 1912, Pe. pinodella (Phoma medicaginis var. pinodella L.K Jones (Mor-37 gan-Jones & K.B. Burch), and Phoma koolunga. For the latter three species, I find that according to the EPPO Global Database and Index Fungorum, the preferred/current names are Didymella pinodes (Berk. & A. Bloxam) Petr., Annls mycol. 22(1/2): 16 (1924); Didymella pinodella (L.K. Jones) Qian Chen & L. Cai, in Chen, Jiang, Zhang, Cai & Crous, Stud. Mycol. 82: 178 (2015); Ascochyta koolunga (J.A. Davidson, D. Hartley, Priest, S. Kaczm., Herdina, A. McKay & E.S. Scott) L.W. Hou, L. Cai & Crous, in Hou, Groenewald, Pfenning, Yarden, Crous & Cai, Stud. Mycol. 96: 381 (2020). For the sake of clarity, I would suggest following EPPO and Index Fungorum taxonomy, unless you have a particular reason for not doing so.

I generally miss information regarding non-inoculated control treatment (which in this case would best be represented by a treatment with water-tween only). I think this must be included to prove that results and symptoms are caused by the pathogens in the inoculum, and not by random contaminant etc.

In the M&M section 2.2 you say that the two fungal species were confirmed as a pure culture of the species by PCR using the species-specific primers (ref 8) and sequencing.  I cannot find any mention of species-specific primers in the literature that you refer to (ref. no 8). I also miss mention of the species identification in the results. Please elaborate.

Expression of PR4 in detached leaves: Was this investigated in detached inoculated leaves, or detached non-inoculated leaves?    

I suggest commenting (in the discussion) on the two species that were not included in the current study (pinodella/koolunga): Whether these could be expected to become a problem, whether any information on species specific vs non-specific resistant mechanisms is available for these species etc.

Minor

Line 4: ….. Kevin, McPhee….  I suppose that the comma should be removed.

Line 125: «…sprayed inoculated… pipetted inoculated…». Instead write … spray inoculated … pipette inoculated

Line 220: include information on how plants were inoculated

Line 230: include information on how plants were inoculated.

Line 230:  “…. And infection type at 9 dpi…” Do you mean that infection type was assessed at 9 dpi?  

Line 231: Same as above

Line 241: include information on when plants were assessed

Line 249 (fig. 5): What does the stippled lines mean?

Line 250: Include info on when plants were assessed

Line 275 (fig. 8): What does the stippled lines mean?

Line 276: Include info on when plants were assessed

I suggest to check all figure headings, and include information on when plants were assessed for symptoms where this is applicable.

Author Response

We appreciate the comments from the two reviewers that helped to improve the manuscript. Our responses to each question of the reviewer are followed.

The manuscript generally reads well, pictures are nice and helpful, and relevant for this important topic.

You use the following taxonomy: Ascochyta pisi Lib. (Didymella pisi sp. nov), Peyronellaea pinodes 35 (Berk & A. Bloxam) Aveskamp, Gruyter & Verkley 2010 (Mycosphaerella pinodes (Berk & 36 A. Bloxam) Vestergr. 1912, Pe. pinodella (Phoma medicaginis var. pinodella L.K Jones (Mor-37 gan-Jones & K.B. Burch), and Phoma koolunga. For the latter three species, I find that according to the EPPO Global Database and Index Fungorum, the preferred/current names are Didymella pinodes (Berk. & A. Bloxam) Petr., Annls mycol. 22(1/2): 16 (1924); Didymella pinodella (L.K. Jones) Qian Chen & L. Cai, in Chen, Jiang, Zhang, Cai & Crous, Stud. Mycol. 82: 178 (2015); Ascochyta koolunga (J.A. Davidson, D. Hartley, Priest, S. Kaczm., Herdina, A. McKay & E.S. Scott) L.W. Hou, L. Cai & Crous, in Hou, Groenewald, Pfenning, Yarden, Crous & Cai, Stud. Mycol. 96: 381 (2020). For the sake of clarity, I would suggest following EPPO and Index Fungorum taxonomy, unless you have a particular reason for not doing so.

--Response: Good point! We revised the paragraph following the EPPO and Index Fungorum taxonomy.

I generally miss information regarding non-inoculated control treatment (which in this case would best be represented by a treatment with water-tween only). I think this must be included to prove that results and symptoms are caused by the pathogens in the inoculum, and not by random contaminant etc.

--Response: Thank you for the concern.  As stated in the M&M section line 92 that the inocula were prepared “with sterilized ddH2O2.”. We did not present a non-pathogen control because the same ddH2O2 did not cause any symptoms on detached leaves and A. pisi-inoculated Fallon plants, so we are confident that the water does not contain any random contaminant.

In the M&M section 2.2 you say that the two fungal species were confirmed as a pure culture of the species by PCR using the species-specific primers (ref 8) and sequencing.  I cannot find any mention of species-specific primers in the literature that you refer to (ref. no 8). I also miss mention of the species identification in the results. Please elaborate.

--Response: Now Ref 8 is a paper submitted to “Plant Disease” recently.

Nyamesorto, B., Johnson, M., Gunnink Troth, E., Parikh, L. P., Crutcher, F. K., Owati, A. S., Agindotan, B., and Burrows, M. E., (2023). Development and Applications of Molecular Assays Specific to Ascochyta pisi, Didymella pinodella, and Didymella pinodes Associated with Ascochyta Blight of Dry Pea Seeds [Manuscript submitted for publication]. Department of Plant Sciences and Plant Pathology, Montana State University.

Expression of PR4 in detached leaves: Was this investigated in detached inoculated leaves, or detached non-inoculated leaves?    

--Response: Expressions of PR4 were investigated in detached non-inoculated leaves.

I suggest commenting (in the discussion) on the two species that were not included in the current study (pinodella/koolunga): Whether these could be expected to become a problem, whether any information on species specific vs non-specific resistant mechanisms is available for these species etc.

--Response: Very good point! We don’t have A. pinodella and A. koolunga in the lab, and A. koolunga has not been reported in the Great Plains of North America We added some comments in the discussion.

Minor

Line 4: ….. Kevin, McPhee….  I suppose that the comma should be removed.

Line 125: «…sprayed inoculated… pipetted inoculated…». Instead write … spray inoculated … pipette inoculated

Line 220: include information on how plants were inoculated

Line 230: include information on how plants were inoculated.

Line 230:  “…. And infection type at 9 dpi…” Do you mean that infection type was assessed at 9 dpi?  

--Response: “An infection type at 9 dpi” meaning the plant in the figure was inoculated 9 days before. We assessed infected plants every day post-inoculation. The dpi below each photo indicated the time point when the photo was taken.

Line 231: Same as above

Line 241: include information on when plants were assessed

Line 249 (fig. 5): What does the stippled lines mean? (?)

Line 250: Include info on when plants were assessed

Line 275 (fig. 8): What does the stippled lines mean?

Line 276: Include info on when plants were assessed

I suggest to check all figure headings, and include information on when plants were assessed for symptoms where this is applicable.

--Response: We appreciated the reviewer’s comments and the information has been added to each figure accordingly.
